# 13*R*,20-Dihydroxydocosahexaenoic Acid, a Novel Dihydroxy- DHA Derivative, Inhibits Breast Cancer Stemness through Regulation of the Stat3/IL-6 Signaling Pathway by Inducing ROS Production

**DOI:** 10.3390/antiox10030457

**Published:** 2021-03-15

**Authors:** Lifang Wang, Hack Sun Choi, Binna Lee, Jong Hyun Choi, Yong-Suk Jang, Jeong-Woo Seo

**Affiliations:** 1Microbial Biotechnology Research Center, Korea Research Institute of Bioscience and Biotechnology (KRIBB), Jeongeup-Si 56212, Korea; zmz0@kribb.re.kr (L.W.); skanwodd@kribb.re.kr (B.L.); jhchoi@kribb.re.kr (J.H.C.); 2Department of Bioactive Material Sciences and the Institute for Molecular Biology and Genetics, Jeonbuk National University, Jeonju 54896, Korea; 3Faculty of Biotechnology, College of Applied Life Sciences, Jeju National University, Jeju 63243, Korea; choix074@jejunu.ac.kr

**Keywords:** breast cancer stem cells, 13*R*,20-dihydroxydocosahexaenoic acid (13*R*,20-diHDHA), mammospheres, ROS, Stat3

## Abstract

Breast cancer is a major health problem worldwide. Cancer stem cells (CSCs) are known to mediate breast cancer metastasis and recurrence and are therefore a promising therapeutic target. In this study, we investigated the anti-inflammatory effect of 13R,20-dihydroxydocosahexaenoic acid (13*R*,20-diHDHA), a novel dihydroxy-DHA derivative, which was synthesized through an enzymatic reaction using cyanobacterial lipoxygenase. We found that 13*R*,20-diHDHA reduced the macrophage secretion of the inflammatory cytokines, IL-6 and TNF-α, and thus appeared to have anti-inflammatory effects. As the inflammatory tumor microenvironment is largely devoted to supporting the cancer stemness of breast cancer cells, we investigated the effect of 13R,20-diHDHA on breast cancer stemness. Indeed, 13*R*,20-diHDHA effectively inhibited breast cancer stemness, as evidenced by its ability to dose-dependently inhibit the mammospheres formation, colony formation, migration, and invasion of breast CSCs. 13*R*,20-diHDHA reduced the populations of CD44^high^/CD24^low^ and aldehyde dehydrogenase (ALDH)-positive cells and the expression levels of the cancer stemness-related self-renewal genes, Nanog, Sox2, Oct4, c-Myc, and CD44. 13R,20-diHDHA increased reactive oxygen species (ROS) production, and the generated ROS reduced the phosphorylation of nuclear signal transducer and activator of transcription 3 (Stat3) and the secretion of IL-6 by mammospheres. These data collectively suggest that 13*R*,20-diHDHA inhibits breast cancer stemness through ROS production and downstream regulation of Stat3/IL-6 signaling, and thus might be developed as an anti-cancer agent acting against CSCs.

## 1. Introduction

Breast cancer is the most common endocrine cancer and the second-leading cause of cancer-related death among women [1,2]. Globally, 15–20% of female breast cancer patients are diagnosed with triple negative breast cancer (TNBC), which is clinically characterized by a high risk of recurrence, metastasis, and short progression-free survival [3,4]. Cancer stem cells (CSCs) are the “seed” of cancer cells and TNBC has a significantly higher proportion of CSCs than other breast cancer subtypes [5,6]. TNBC uses unique mechanisms and aspects of its tumor microenvironment (TME) to maintain its CSC phenotype, which critically contributes to chemotherapy resistance, metastasis, and tumor recurrence [7]. Most of the available targeted and chemotherapeutic drugs target tumorigenic mutations and/or kill highly proliferative cancer cells. They often miss quiescent CSCs, and surviving CSCs can generate new (sometimes drug-resistant) cancer cells, thereby promoting tumor progression [8,9]. In recent years, research into CSCs has led to CSC-targeting therapeutic strategies that have shown efficacy in preclinical studies of TNBC, mainly by blocking the key molecules that maintain the phenotype of breast CSCs, changing the TME, and/or enhancing the sensitivity of CSCs to chemotherapeutic drugs [10,11]. However, we still lack a complete understanding of breast CSCs, their regulatory mechanisms, and their targets. This has greatly impeded the development and application of relevant therapeutic strategies.

Breast CSCs are characterized by the properties of self-renewal and transformation between the states of being relatively static, invasive, stromal, and epithelioid (more proliferative) [12]. The self-renewal of CSCs is crucially modulated by the activations of particular cell signaling pathways, such as those involving nuclear transcription factor-κB (NF-κB), Stat3, PI3K/AKT/MAPK, Sonic hedgehog (Shh), Wnt/β-catenin, TGF-β, and Notch [13,14,15]. Activation of inflammatory signaling pathways in breast cancer cells induces increases in NF-κB and Stat3 activity, which can drive the formation of breast CSCs. Under an inflammatory TME, NF-κB and Stat3 are activated to stimulate the production of the cytokines, IL-6 and IL-8 [16]. IL-6 can regulate the crosstalk between CSCs and cancer cells through constitutive activation of Stat3, and Stat3 regulates the expression of Oct3/4, which is a major reprogramming factor known to induce the expression of various stemness genes [16,17]. Breast CSCs also express specific surface markers and proteins, including CD44, CD133, Oct4, Sox2, Nanog, and enzyme aldehyde dehydrogenase (ALDH). The CD44^high^/CD24^low^ cell population has been associated with breast CSC characteristics [18,19]. These stemness markers of CSCs can be reduced by intracellular and extracellular factors. Many studies have indicated that ROS also has an impact on stem cell differentiation, with ROS exposure causing stem cells to lose their “stemness” and die.

ROS represent a class of oxygen reduction-derived products mainly comprising free radicals (O_2_^●^^−^, HO^●^, NO^●^, ONOO^−^, etc.) and peroxides that easily form free radicals (H_2_O_2_, O_3_, NO^●^_2_, and HOCl). Intracellular ROS can be produced by mitochondria, NADPH oxidase complex, CytoP450, COX, peroxisomes, and the endoplasmic reticulum. Cancer cells have a higher redox state than normal cells, and this has vital meaning for the proliferation, infiltration, metastasis, and chemo/radio-resistance of cancer cells [20,21,22,23,24]. However, ROS is a double-edged sword: On the one hand, moderately elevated ROS can promote tumor cell growth and proliferation by regulating ROS-related signaling pathways, such as the growth factor and receptor tyrosine kinase pathways; on the other hand, excessive ROS induced by cancer therapy drugs can cause oxidative stress, DNA damage, and the accumulation of oxidized lipids and proteins, ultimately inducing cell cycle arrest, apoptosis, and necrosis [25,26]. CSCs tend to have lower ROS levels than differentiated cancer cells, and these low amounts of ROS are actually needed to maintain the quiescence and self-renewal potential of CSCs [27,28]. It has thus been proposed that increased ROS could contribute to reducing the stemness and enhancing the differentiation of various stem cells. Sato’s research group reported that H_2_O_2_ inhibits self-renewal and induces differentiation of glioblastoma stem cells by increasing intracellular ROS without substantially reducing viability [29]. Zhao’s research group reported that exposure of liver cancer stem cells to exogenous ROS or elevating endogenous ROS by inhibiting the cellular antioxidant mechanism could promote the endothelial differentiation of these cells [30]. ROS generation has been reported to be a causal factor in promoting the differentiation of human mesenchymal stem cells to adipocytes, and ROS levels were reported to increase during the adipogenic differentiation of human adipose-derived stem cells [31,32]. It has also been documented that generation of mitochondrial ROS is progressively increased during hematopoietic stem cell differentiation, and that this occurs via targeting of Notch signaling and is regulated by the autophagy pathway [33]. These results indicate that ROS plays an important role in regulating differentiation in stem cells, especially CSCs.

Excessive and uncontrolled inflammation is now recognized as an underlying component in chronic diseases and even cancers [34]. The ideal response to inflammation in humans is a self-limited inflammatory response leading to complete resolution [35]. The resolution phase is now widely recognized as a biosynthetically active process, governed by a superfamily of endogenous chemical mediators, namely specialized pro-resolving mediators (SPMs), include resolvins, protectins, and maresins [36]. Inflammatory TME is a major component of the cancer cell stemness, metastasis, relapse, and negative outcomes of cancer patients. Thus, resolution of inflammation is crucial in efforts to reduce the cancer stemness of CSCs. Among resolvins, the benefits of resolvin Ds have been reported in various cancers, including those of the lung, liver, pancreas, stomach, and colon [37,38,39,40,41,42,43,44,45,46,47]. Resolvin D1 reportedly inhibits hyper-expressed c-Myc by attenuating its phosphorylation-dependent stabilization in HCT116 colon cancer cells, and has also been shown to prevent the progression of hepatitis to liver cancer [40,48]. Resolvin D1 and resolvin D2 at low concentrations (100 pM) were shown to inhibit the adherence and proliferation of a human oral squamous cell carcinoma cell line (HSC-3) in vitro [49]. Resolvin D1, resolvin D2, and resolvin E1 were found to inhibit the debris-stimulated inflammatory prostate tumor, breast tumor, and liver tumor, etc. [46]. The anti-tumor activities and mechanisms of resolvin differ by the cancer type, and a previous study suggested that the anti-tumor activity of resolvin is mediated through stromal cells instead of a direct action on tumor cells. This was speculated to occur through mechanisms such as enhancing the clearance of debris via macrophage-mediated phagocytosis; modulating the macrophage polarization of M1 type, M2 type, and tumor-associated macrophages; and by reducing the number of cancer mediator-induced CD22b^+^Ly6G^−^ myeloid cells [46,49,50]. Resolvin D1 prevents epithelial mesenchymal transition (EMT) and reduces the stemness of hepatocellular carcinoma (HCC) by inhibiting the paracrine action of cancer-associated fibroblast (CAF) [51]. Resolvin D1 was reported to inhibit EMT in A549 lung cancer cells, and aspirin-triggered resolvin D1 was shown to decrease EMT through inhibition of the mTOR pathway, which is closely linked to oxidative stress [41,51,52]. However, the anti-CSC activity of resolvins had not previously been examined.

13*R*,20-diHDHA is a novel resolvin that we previously synthesized by site-directed mutagenesis of lipoxygenase derived from *Oscillatoria nigro-viridis PCC 7112* grown with docosahexaenoic acid (DHA) as the substrate. Its structure was investigated by LC-MS/MS and NMR in our previous work [53]. In the present study, we further demonstrate that 13*R*,20-diHDHA has potential anti-inflammatory effects and can inhibit breast cancer stemness by inducing ROS production to alter Stat3/IL-6 signaling, and thus may be a promising potential therapeutic agent acting against breast CSCs.

## 2. Materials and Methods

### 2.1. Materials

13*R*,20-diHDHA [53] (purity > 98%) was purified and obtained from DHA through an enzymatic reaction using the cyanobacterial lipoxygenase, as previously described. Cell growth was assessed using a CellTiter 96^®^ AQueous One Solution kit (Promega, Madison, WI, USA). An ALDEFLUOR™ kit was obtained from Stemcell Technologies, Inc. (Vancouver, BC, Canada) and used for ALDH activity determination. Chemicals such as N-acetylcysteine (NAC), phorbol 12-myristate 13-acetate (PMA), and lipopolysaccharide (LPS) were obtained from Sigma-Aldrich (St. Louis, MO, USA). 13R,20-diHDHA was stored at −20 °C in 100% dimethyl sulfoxide (DMSO). The final DMSO concentration was <0.1% and the control group was treated with DMSO alone. A human monocytic cell line (THP-1) and human breast cancer cell lines (MDA-MB-231 and MCF-7) were purchased from the Korea Cell Line Bank (KCLB, Seoul, Korea).

### 2.2. IL-6 and TNF-α Cytokine Determination

PMA-differentiated THP1 macrophages were subjected to LPS-induced inflammation according to the previously published protocol [54]. Briefly, 100 μL of THP1 cell suspension containing 2 × 10^5^ cells was seeded to each well of a 96-well-plate, and the THP1 cells were differentiated to macrophages by exposure to PMA (10 ng/mL) for 72 h. The cells were washed three times with PBS, rested overnight, and then stimulated by 1 μg/mL LPS and treated with or without 13R,20-diHDHA at various concentrations. After 48 h, 50 μL of supernatant was collected and centrifuged at 1000 rpm for 5 min, and the secreted levels of IL6 and TNF-α were tested using ELISA kits (Abcam, Cambridge, UK).

### 2.3. Cell Culture and Mammospheres Formation and Colony Formation

MCF-7 and MDA-MB-231 breast cancer cells were grown in RPMI1640 medium supplemented with 10% (*v*/*v*) fetal bovine serum (FBS; HyClone, Logan, UT, USA) and 1% penicillin/streptomycin (Gibco, Carlsbad, CA, USA) in T75 culture flasks, and incubated in a humidified incubator with 5% CO_2_ at 37 °C. For mammospheres formation, single-cell suspensions of cancer cells were seeded at a density of 4 × 10^4^ (MCF-7) or 1 × 10^4^ cells (MDA-MB-231)/well in ultralow attachment 6-well plates with 2.5 mL/well of complete MammoCult^TM^ Medium (Stemcell Technologies, Vancouver, BC, Canada) supplemented with heparin (4 μg/mL), hydrocortisone (0.48 μg/mL) and 13R,20-diHDHA, then incubated for 7 days. For counting of mammospheres, we followed the method reported by Kim [55]. Briefly, 6-well-plate was scanned by a scanner (Umax PowerLook 1100; Lasersoft Imaging, Seoul, Korea), then images were analyzed by NICE software program. Mammospheres formation efficiency (MFE, %) was measured using the formula: (number of mammospheres observed in control or drug-treated cultures/number of spheres observed in the DMSO control × 100). For colony formation, two cell lines were seeded at a low density in a 6-well plate, treated with 13*R*,20-diHDHA (20 or 40 μM) in medium, and incubated for 7 days. The grown colonies were fixed with 3.7% formaldehyde for 10 min, stained with 0.05% crystal violet for 30 min, and washed by PBS for three time, then imaged by the scanner.

### 2.4. Cell Proliferation

We used a CellTiter 96^®^ Aqueous One Solution assay kit (Promega) to measure the proliferation rates of MCF-7 and MDA-MB-231 cells [55]. The two cell lines were seeded at 1.5 × 10^4^ cells per well in a 96-well plate for 24 h and then incubated with increasing concentrations of 13R,20-diHDHA (10, 20, 40, and 80 μM) for 24 h. Proliferation was measured according to the manufacturer’s protocol, and the optical density (OD_490_) was determined with a microplate reader (Biotek, Seoul, Korea).

### 2.5. Annexin V/Propidium Iodide (PI) Assay for Apoptosis and Hoechst 33,342 Staining of Apoptotic Nuclei

MDA-MB-231 cells and MCF-7 cells were cultured in 6-well plates with 13R,20-diHDHA (20 or 40 μM) for 24 h. The cells were washed with cold PBS, resuspended in 1× binding buffer, and treated with 5 μL of FITC Annexin V and 5 μL of PI. The cells were gently vortexed, and incubated for 15 min at RT (25 °C) in the dark. Finally, we added 400 μL of 1× binding buffer and analyzed the samples with a FACS system (BD, San Jose, CA, USA).

### 2.6. Scratch Assay

MDA-MB-231 cancer cells were inoculated to a 6-well plate at 2 × 10^6^ cells/well, allowed to adhere overnight, and then grown to a monolayer. A scratch was made in the monolayer using a 200 μL tip. The plates were washed twice with 1 × PBS, and 13R,20-diHDHA (20 or 40 μM) was added in fresh RPMI1640/0.5% FBS for 12 h. Photomicrographs of the wounded areas were acquired by a light microscope. The cells that migrated across the wounds were counted in nine randomly chosen fields from each triplicate treatment. The percentage of inhibition was expressed relative to the migration seen in control group, which was taken as 100%.

### 2.7. Transwell Assay

We performed the transwell assay according to a previously described method [56]. Briefly, 8-μm-pore polycarbonate membranes (Merck, Millipore, Darmstadt, Germany) coated with/without a Matrigel matrix basement (BD) in 24-well hanging inserts were used for invasion and migration assays. The upper chamber was loaded with 200 μL of MDA-MB-231 suspensions (1 × 10^5^ cells) treated with 20 μM or 40 μM 13R,20-diHDHA in RPMI1640 supplemented with 0.5% FBS, the bottom chamber was loaded with 900 μL RPMI1640 containing 20% FBS, and the cells were cultured for 48 h. The cells that passed through the membrane were fixed with 4% paraformaldehyde and stained with 0.03% crystal violet. Images were acquired using an inverted light microscope.

### 2.8. Flow Cytometric Analysis of CD44^high^/CD24^low^ Cells and ALDH Activity

Expression of CD44 and CD24 was determined by FACS analysis in MDA-MB-231 and MCF-7 cells as previously described by Zhen [56]. One million cells were suspended, treated with 13R,20-diHDHA (20 or 40 μM) for 48 h, harvested with 1 × trypsin/EDTA, washed with 1 × PBS, and labeled with FITC-conjugated anti-human CD44 and PE-conjugated anti-human CD24 antibodies (BD Pharmingen, San Diego, CA, USA). The cells were washed twice with FACS buffer and analyzed by FACS. An ALDEFLUOR™ kit (Vancouver, BC, Canada) was used to measure the ALDH activity. The cancer cells were treated with 13R,20-diHDHA (20 or 40 μM) for 24 h and incubated in ALDH assay buffer at 37 °C for 30 min. The ALDH inhibitor, diethylaminobenzaldehyde (DEAB), was used as a negative control, and ALDH-positive and ALDH-negative cells were assayed using FACS.

### 2.9. Measurement of ROS Activity Using DCFDA (2′,7′-Dichlorofluorescein Diacetate) Probe Detection Method

Cancer cells in 96-well cell imaging plates and treated with 13R,20-diHDHA (20 or 40 μM) or DMSO for 24 h, and ROS was detected using DCFDA. Briefly, the cells were washed with 1 × PBS and incubated with PBS containing 10 μM DCFDA probe for 30 min at room temperature, then cells were washed with 1 × PBS. Finally, 0.1 mL PBS was added to each well, and the samples were observed under a phase-contrast fluorescence microscope (Lionheart FX live cell imager; Biotek, Winooski, VT, USA) and analyzed by FACS.

### 2.10. Quantitative Measurement of Human Cytokines

Human cytokines were measured using a human inflammatory cytokine assay kit (BD, San Diego, CA, USA). MDA-MB-231 cell mammospheres were seeded in ultralow attachment 6-well plates containing 2 mL of complete MammoCult™ medium for 5 days and incubated with 13R,20-diHDHA (20 µM) for 2 days. We then performed the above mentioned assay as described by the manufacturer. For detection of secreted IL-6 levels, 50 µL of mixed capture beads, 50 µL of culture or standard medium, and 50 µL of PE detection reagent were added to each assay tube. The samples were incubated at room temperature for 3 h with protection from light, washed with a washing buffer, and centrifuged. Each beads pellet was washed and resuspended in 300 µL of washing buffer, and the samples were analyzed using FACS.

### 2.11. Gene Expression Analysis

Total RNA was isolated using TaKaRa MiniBEST (TaKaRa, Tokyo, Japan) according to the supplier’s protocol. The levels of transcripts also were determined by a One step AccuPower GreenStar^TM^ RT-qPCR PreMix kit using SYBR Green according to the manufacturer’s instructions (Bioneer Corporation, Daejeon, Korea). RT-PCR was carried out using 100 ng of total RNA, a reaction volume of 50 μL, and the specific primers listed in Table 1. The PCR cycle conditions consisted of 95 °C for 0.5 min, 60 °C for 0.5 min, and 72 °C for 0.5 min, followed by a 10-min extension at 72 °C. The relative mRNA expression levels of the target genes were calculated using the comparative CT method. At least four independent PCR procedures were performed to allow for statistical analysis. The PCR product levels obtained were normalized to that of the β-actin gene as an internal control.

### 2.12. Western Blot Analysis

Mammospheres were treated with and without 13R,20-diHDHA (20 μM) and lysed with lysis buffer containing 1 mM phenylmethanesulfonyl (PMSF) and a proteinase inhibitor cocktail on ice for 45 min, and the samples were centrifuged at 12,000 rpm for 5 min. The isolated proteins were separated by 10% or 12% SDS-PAGE and transferred to a polyvinylidene difluoride membrane (PVDF; Millipore, Bedford, MA, USA). Each PVDF membrane was blocked in 5% skim milk in Tris-buffered saline/Tween 20 (0.1%, *v*/*v*; TBST) at room temperature for 1 h and then incubated with primary antibodies at 4 °C overnight. The primary antibodies used were all obtained from (Abcam, Cambridge, MA, USA) and included anti-cleaved-caspase-3 (ab2302), anti-Stat3 (ab68153), anti-pStat3 (ab76315), anti-p65 (ab16502), anti-IL-6 (CST#12912), anti-β-actin (ab8227), and anti-Lamin B1 (ab16048). Each PVDF membrane was washed three times with TBST, incubated with HRP-conjugated secondary antibodies (ab205718), developed using a chemiluminescence detection kit (BioRad, Hercules, CA, USA), and imaged with a chemi-doc machine (iBright CL1500 imaging system; ThermoFisher Scientific, Waltham MA, USA). Densitometric analysis of the Western blot data was performed using the iBright analysis program.

### 2.13. Statistical Analysis

All data are presented as the mean ± standard deviation (SD). Data were analyzed using the Student’s *t*-test. A *p* value lower than 0.05 was considered statistically significant. All analyses were performed using the GraphPad Prism 8 Software (GraphPad Software Inc., San Diego, CA, USA).

## 3. Results

### 3.1. Effect of 13R,20-diHDHA on IL-6 and TNF-α Cytokine Secretion

13*R*,20-diHDHA is a novel resolvin that we previously synthesized by site-directed mutagenesis of lipoxygenase derived from *Oscillatoria nigro-viridis PCC* 7112 [53]. The 13*R*,20-diHDHA isolation method is summarized in Figure 1A. Briefly, recombinant lipoxygenase was purified, its specific activity was determined, and it was used to convert DHA to hydroxy fatty acids. The converted products were refined using HP20 resin, and the final structure was determined by HPLC, LS-MS/MS, and NMR analyses. To elucidate the anti-inflammatory effect of 13*R*,20-diHDHA, we used LPS to induce THP1 macrophage cells with or without 13*R*,20-diHDHA, then detected the secretion of IL-6 and TNF-α by ELISA. As shown in Figure 1B, we observed a significant reduction of IL-6 and TNF-α secretion in samples treated with 1 ppM 13R,20-diHDHA for 48 h. Analysis of various doses showed that this inhibition was 13*R*,20-diHDHA dose-dependent across a concentration range of 1–100 ppM. This result shows that 13*R*,20-diHDHA has as strong an anti-inflammatory effect.

### 3.2. Effect of 13R,20-diHDHA on Cell Viability and Mammospheres Formation in Breast Cancer Cells

To evaluate whether 13*R*,20-diHDHA has a potent anti-cancer effect, we first tested the cell viability of MCF-7 and MDA-MB-231 cells exposed to 13R,20-diHDHA at various concentrations. As shown in Figure 2A, 13*R*,20-diHDHA did not show any inhibitory activity on breast cancer cell viability. This is not unexpected, because the anti-tumor activity of resolvin Ds differ across cancer types and has been speculated to be at least in some cases mediated by the TME rather than a direct action on tumor cells [46]. Thus, to further investigate whether 13*R*,20-diHDHA can suppress the formation of breast cancer cells mammospheres, we treated MCF-7 and MDA-MB-231 cell-derived mammospheres with various concentrations of 13*R*,20-diHDHA. Our results showed that 13R,20-diHDHA at 20 μM decreased the mammospheres number by 54% and reduced the size of the formed mammospheres (Figure 2B, Appendix A). To assess whether 13*R*,20-diHDHA regulates breast cancer mammospheres growth, we applied 13*R*,20-diHDHA to mammospheres and counted cancer cell numbers. As shown in Figure 2C, the cancer cell number was decreased in 13R,20-diHDHA-treated mammospheres, indicating that 13*R*,20-diHDHA treatment decreases mammospheres cell proliferation. As shown in Figure 2D and Appendix A, Annexin-V-PI double staining showed that the apoptotic marker AnnexinV/PI positive population was slightly but not significantly increased by 13*R*,20-diHDHA treatment of breast cancer cells, from 0.6% in the control group to 2.9% in the 40 μM 13R,20-diHDHA-treated group. As shown in Figure 2E, there was no significant difference in the level of the apoptotic marker, cleaved caspase 3, in breast CSC mammospheres treated with or without 13R,20-diHDHA. These results indicate that 13*R*,20-diHDHA does not directly induce the apoptosis of breast cancer cells in cultures or cultured mammospheres. However, additional experiments revealed that treatment with 13*R*,20-diHDHA suppressed the migration, invasion, and colony formation of breast cancer cells (Figure 2F–H, Appendix A). These results collectively show that 13*R*,20-diHDHA can inhibit the mammospheres formation, colony formation, migration, and invasion of breast cancer cells lines, and thus has the potential to be an anti-CSC agent.

### 3.3. 13R,20-diHDHA Reduces the CD44^high^/CD24^low^-Expressing and ALDH-Positive Cancer Cell Populations

The surface marker proteins used extensively to identify cancer stem cells include the transmembrane glycoprotein, CD44, the cell adhesion protein, CD24, and the cytosolic enzyme, ALDH-1, which oxidizes aldehydes to carboxylic acids [57]. The subpopulation of breast cancer expressing CD44^high^/CD24^low^ in clinical specimens had a high capacity to form tumors [58]. Here, we investigated the CD44^high^/CD24^low^ and ALDH-positive populations of breast cancer cells after 13*R*,20-diHDHA treatment. Our results showed that 13*R*,20-diHDHA reduced the CD44^high^/CD24^low^ cell proportion from 75.1% to 44.8% (Figure 3A, Appendix A) and the ALDH-positive cell fraction from 1.7% to 0.4% (Figure 3B, Appendix A). To test whether 13*R*,20-diHDHA regulates the transcript levels of self-renewal genes, we used real-time RT-qPCR. Our results showed that 13*R*,20-diHDHA reduced the transcript levels of the self-renewal genes, Nanog, Sox2, Oct4, c-Myc, and CD44, in CSCs (Figure 3C). These findings illustrate that 13*R*,20-diHDHA effectively reduces the stemness of breast CSCs.

### 3.4. 13R,20-diHDHA Induces ROS Generation, and NAC Reverses 13R,20-diHDHA-Induced Mammospheres Inhibition

In general, increased ROS kills CSCs, whereas low levels of ROS are associated with the stemness of stem cells and CSCs [59]. In numerous cancer types, persistently upregulated ROS-dependent signaling pathways have been implicated in cell differentiation, growth, and survival [60,61]. To explore the mechanism through which 13R,20-diHDHA reduced the stemness of breast CSCs, we measured ROS production of CSCs treated with/without 13R,20-diHDHA using the DCFDA probe. Results obtained from fluorescence microscopy and FACS showed that 13*R*,20-diHDHA treatment increased the production of ROS in CSCs without altering their cell viability (Figure 4A,B). We usually thought excessive ROS can cause oxidative stress, DNA damage, and the accumulation of oxidized lipids and proteins, ultimately inducing cell cycle arrest, apoptosis, and necrosis, however, lactic acidosis and L-buthionine sulfoximine induces a much higher cellular ROS level but permits a progressive growth of the tested cancer cells such as: 4T1, 4T1, Bcap37, RKO, SGC7901 [25,26,62], which is consistent with our research results. It has thus been proposed that increased ROS could contribute to reducing the stemness and enhancing the differentiation of various stem cells. Sato’s research group reported that H_2_O_2_ inhibits self-renewal and induces differentiation of glioblastoma stem cells by increasing intracellular ROS without substantially reducing viability [29]. Thus, we hypothesized that 13*R*,20-diHDHA-induced ROS could regulate the differentiation of CSCs and reduced their stemness, drug-resistance, tumor angiogenesis, and metastasis. To address this possibility, we tested the effect of the ROS inhibitor, N-acetyl-L-cysteine (NAC), on mammospheres formation in our system. Indeed, our results showed that NAC reversed the 13*R*,20-diHDHA-induced reduction in the MFE of MDA-MB-213 and MCF-7 cells (Figure 4C, Appendix A).

### 3.5. 13R,20-diHDHA Inhibits Breast CSCs Stemness through Stat3/IL-6 Signaling

The ROS-mediated activation of the Stat3 signaling pathway was previously reported to be involved in cellular senescence [63], and NF-kB activity is essential for maintaining the survival of breast CSCs [64]. To determine the biological function of 13R,20-diHDHA, we examined the Stat3 and NF-κB pathways in mammospheres derived from MDA-MB-231 and MCF-7 cells under 20 μM 13*R*,20-diHDHA treatment. As shown in Figure 5A,B, 13*R*,20-diHDHA reduced the levels of nuclear pStat3 and p65 protein compared to those in the control group, and NAC reversed the 13*R*,20-diHDHA-induced dephosphorylation of Stat3 but not p65. NF-κB and Stat3 are activated to stimulate the production of the cytokines, IL-6 and IL-8 [16]. IL-6 can regulate the crosstalk between CSCs and cancer cells through constitutive activation of Stat3, and plays an important role in the formation of mammospheres [65]. To determine the level of secreted IL-6, we performed FACS analysis of mammospheres-cultured supernatants by human inflammatory cytokines assay kit (BD, San Diego, CA, USA). As shown in Figure 5C,D, 13*R*,20-diHDHA reduced the expression and secretion of IL-6 in our system, and NAC reversed the reduction of IL-6 secretion.

## 4. Discussion

Although tremendous progress has been made in encouraging women to undergo regular screenings, identifying breast lesions at earlier stages, and developing an array of combination therapeutic strategies, breast cancer remains the second leading cause of cancer-related mortality in women [66,67]. The major hurdles include poor de novo response to therapies; development of acquired resistance leading to recurrent disease and metastasis; and non-compliance in patients owing to poor tolerance of drug-related side effects [68]. Some tumor cells may undergo mutations and epigenetic alterations in their signaling pathways and/or generate an inflammatory TME, which can lead to the formation of CSCs. Such CSCs can undergo self-renewal and differentiation, similar to normal stem cells, and contribute to the drug resistance of breast cancer patients [69]. Although several therapeutic methods have been designed to target CSCs, it is still unclear how to efficiently target breast CSCs. In theory, this could be done directly and/or by interrupting the inflammatory status of the TME [70].

Resolvins and their precursors have been shown to exhibit anti-tumor activities in various cancers through multiple mechanisms, such as by targeting angiogenesis, EMT, pro-tumorigenic cytokines, natural killer cells, and macrophages. In this study, our results indicate that the novel anti-inflammatory and bioactive resolvin-derived agent, 13R,20-diHDHA, has anti-CSC effects in breast cancer, as follows: (1) 13*R*,20-diHDHA reduces the secretion of IL-6 and TNF-α by LPS-stimulated (inflamed) macrophages (Figure 1). (2) 13*R*,20-diHDHA inhibits the size and formation of MDA-MB-231 cell-derived mammospheres (Figure 2A–D). (3) 13*R*,20-diHDHA inhibits the migration, invasion, and colony formation of breast cancer cells (Figure 2E–H). (4) 13*R*,20-diHDHA reduces the CD44^high^/CD24^low^, ALDH-positive populations breast CSCs and the transcript levels of self-renewal genes in breast cancer cells (Figure 3). (5) 13*R*,20-diHDHA inhibits the mammospheres formation of breast CSCs by increasing ROS production, and this effect can be reversed by NAC (Figure 4A–C). (6) 13*R*,20-diHDHA inhibits the expression and secretion of IL-6 (an important cytokine of CSCs), signaling pathways involving Stat3. Together, these lines of evidence indicate that 13*R*,20-diHDHA may act as an anti-cancer agent in breast cancer by targeting breast CSCs.

This work is part of our group’s efforts to develop novel and economic resolvins with anti-inflammatory effects. 13*R*,20-diHDHA was synthesized by site-directed mutagenesis of lipoxygenase derived from *Oscillatoria nigro-viridis PCC 7112* grown on DHA as a substrate [53]. In recent decades, novel resolvins have been increasingly synthesized, isolated, and identified. In 2010, Lu et al. found that wounding induced formation of a novel endogenous 14,21-diHDHA and macrophage function as the combination of 12-LOX and cytochrome P450 to generate these 14,21-diHDHA stereoisomers and intermediates [71]. In 2014, 14,20-diHDHA was biosynthesized by eosinophils through the 12/15-lipoxygenzse pathway, and nanogram doses of synthetic 14,20-diHDHA were found to display a potent anti-inflammatory action by reducing PMN infiltration in zymosan-induced peritonitis [72]. In our previous work, we synthesized 13*R*,20-diHDHA. Here, we verified its anti-inflammatory activity by showing that 13*R*,20-diHDHA reduced the secretion of IL-6 and TNF-α. As the inflammatory TME is crucial for the cancer stemness of breast cancer cells, we further investigated the anti-CSC potential of 13*R*,20-diHDHA.

The mammospheres assay has been widely utilized to measure in vitro stem/progenitor cell frequency in normal primary mammary epithelial cell preparations and the frequency of CSCs or tumor-initiated cells derived from malignant mammary tissue [73,74]. Although the gold-standard stem cell assay is the in vivo transplantation assay, the mammospheres assay offers investigators an in vitro assay that is less time consuming and more cost effective than the in vivo transplantation assay [75,76]. Here, we report that 13*R*,20-diHDHA inhibited the mammospheres formation of breast cancer cells (in both size and number) without obviously decreasing cell viability, although it conferred a slight induction of apoptosis. To assess for changes in diverse biological properties of breast cancer cells under 13*R*,20-diHDHA treatment, we tested colony formation, cell migration, and cell invasion. Our results showed that our novel resolvin significantly inhibited colony formation, cell migration, and cell invasion, and thus could reduce the stemness of breast CSCs. To confirm this, we investigated changes in additional breast CSC markers, and found that 13*R*,20-diHDHA treatment decreased the populations of CD44^high^/CD24^low^ and ALDH-positive cells, and the expression level of the self-renewal genes, CD44, CD133, Oct4, Sox2, and Nanog. 13*R*,20-diHDHA also increased ROS production in breast cancer CSCs without cytotoxicity, which argues against the concept that increasing ROS in cancer cells should kill these cells. However, there is no evident dose-response relationship between the cellular ROS level and cytotoxicity. Our observation that 13*R*,20-diHDHA can increase ROS production is comparable to previous findings that lactic acidosis and L-buthionine sulfoximine can induce ROS levels without negatively impacting the growth of various cancer cell lines [62]. Besides, increased ROS could contribute to reducing the stemness and enhancing the differentiation of stem cells, as Sato’s research group reported that, H_2_O_2_ inhibits self-renewal and induces differentiation of glioblastoma stem cells by increasing intracellular ROS without substantially reducing viability. Based on this, we speculate that the excessive ROS might alter the TME in our system. We found that the ROS inhibitor, NAC, attenuated 13*R*,20-diHDHA-induced ROS generation and reversed the 13*R*,20-diHDHA-induced inhibition of mammospheres.

Another study found that increasing intracellular ROS via the activation of NADPH oxidase complex 5 inhibited breast cancer stemness through the Stat3 signaling pathway in mammospheres [11,55]. Here, we assessed Stat3 signaling after 13*R*,20-diHDHA-induced ROS production in breast CSCs, and found that this treatment decreased Stat3 signaling, IL-6 expression and secretion, and mammospheres formation. The Stat3 signaling pathway is critical for normal stem cell functions and plays an important role in breast CSCs. Our results revealed that 13*R*,20-diHDHA reduced the nuclear translocation of pStat3 and p65, but that 13*R*,20-diHDHA-induced ROS was involved only in the dephosphorylation of pStat3. This is notable because constitutively activated pStat3 is responsible for 30–60% of primary breast cancer [77,78,79]. There are endogenous protein inhibitors of Jak kinases, which include SOCS1 or SOCS3. SOCS proteins are direct target genes of STATs, which inhibit JAK/STAT signaling via a classic negative feedback loop [80]. It is possible that 13*R*,20-diHDHA activates either SOCS1 or SOCS3, which directly inhibits Jak2, thus leading to reduced phosphorylation of STAT3. The decline in STAT3 phosphorylation induced by their compound could also have been caused by activation of a protein tyrosine phosphatase such as SHIP1 or SHIP2, which are known to dephosphorylate STAT3 resulting in negative regulation [81]. Thus, it is also possible that 13*R*,20-diHDHA activated SHIP1 or SHIP2, which dephosphorylated STAT3 downstream of the IL-6R activation. Our manuscript showed that ROS induces STAT3 dephosphorylation in cells treated with our compound.

Together, our data suggest the following proposed model for CSC stemness inhibition by 13*R*,20-diHDHA: 13*R*,20-diHDHA induces ROS production, which induces dephosphorylation of Stat3 to reduce the expression and secretion of IL-6. Secreted IL-6 can convert NSCCs to CSCs and regulates the dynamic equilibrium from NSCCs to CSCs. 13*R*,20-diHDHA deregulates this equilibrium through dephosphorylation of Stat3 and deregulation of IL-6, and thereby reduces the cancer stemness of breast cancer cells. This indicates that 13*R*,20-diHDHA could potentially be developed as a new breast cancer chemo-preventive agent.

## 5. Conclusions

We herein show that 13*R*,20-diHDHA inhibits the mammopshere formation, colony formation, and migration of breast cancer cells. Mechanistically, 13R,20-diHDHA induces ROS production to reduce the nuclear phosphorylation of Stat3 and the secretion of IL-6 by mammospheres. Our results suggest that 13*R*,20-diHDHA deregulates the dynamic equilibrium from non-stem cancer cells to CSCs by dephosphorylating Stat3 and decreasing IL-6 secretion, thereby inhibiting CSC formation. Our results collectively show that 13*R*,20-diHDHA may be a promising potential therapeutic agent acting against breast CSCs.

## Figures and Tables

**Figure 1 antioxidants-10-00457-f001:**
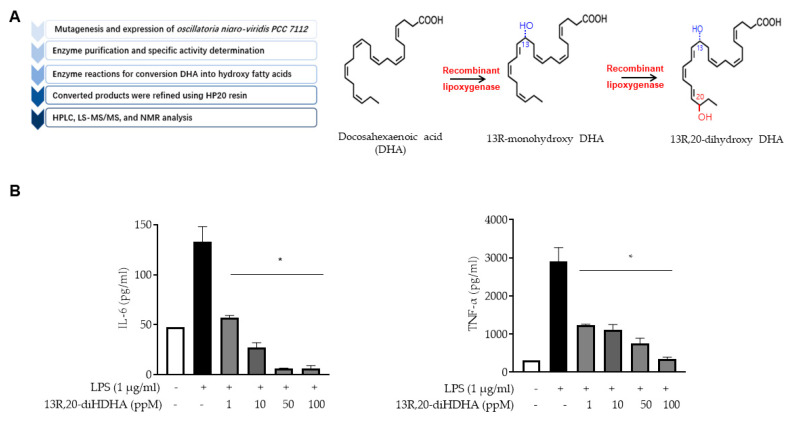
13*R*,20-diHDHA inhibits IL-6 and TNF-α production, as assessed by a cytokine assay. (**A**) Summary of the process used to produce 13R,20-diHDHA. (**B**) The secretions of IL-6 and TNF-α, by LPS-stimulated (inflamed) macrophages were inhibited by various concentrations of 13*R*,20-diHDH. THP1 cells were treated with PMA to generate macrophages, which were stimulated with LPS (1 μg/mL) for 48 h with or without 13*R*,20-diHDHA. Cytokines were measured by ELISA. The data from triplicate experiments are presented as the mean ± SD. * *p* < 0.05 versus the LPS-treated positive control group.

**Figure 2 antioxidants-10-00457-f002:**
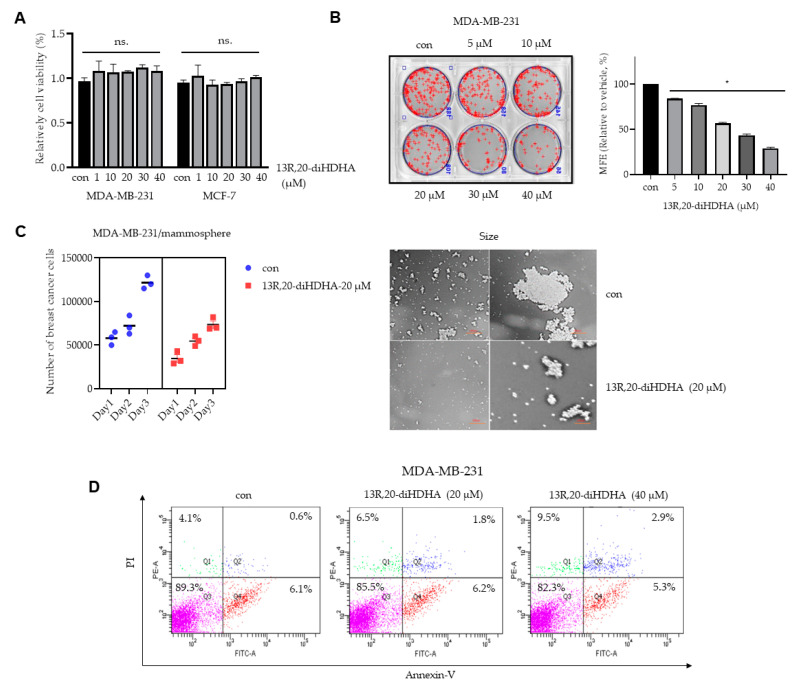
The effect of 13*R*,20-diHDHA on mammospheres formation and multiple cancer hallmarks in breast cancer cell lines. (**A**) MDA-MB-231 and MCF-7 cells were cultured in a 96-well plate with various concentrations of 13*R*,20-diHDHA for 24 h and cancer cell proliferation was assayed with a CellTiter 96^®^ AQueous One Solution kit. (**B**) The mammospheres formation efficiency (MFE) was decreased by 13*R*,20-diHDHA treatment. Mammospheres derived from MDA-MB-231 cells were cultured for 7 days in the presence of 13*R*,20-diHDHA or DMSO. Image shows the sizes of representative mammospheres, as obtained by microscopy (scale bar: 100 μm). * *p* < 0.05 versus the DMSO-treated control group. (**C**) 13*R*,20-diHDHA prevents mammospheres growth. Mammospheres were treated with 13*R*,20-diHDHA for 2 days and dissociated to single cells, then equal numbers of cells were plated to fresh dishes. The cells were counted on days 1,2,3 in triplicate and plotted as the mean value. The data from triplicate experiments are presented as the mean ± SD. (**D**) 13*R*,20-diHDHA does not induce significant apoptosis of MDA-MB-231 cells. Apoptosis was determined using Annexin V/propidium iodide (PI) staining and FACS. (**E**) During the mammospheres formation process, 13*R*,20-diHDHA does not induce a significant change in the level of cleaved caspase 3 (a marker of apoptosis), as determined by western blot analysis. β-actin was used as an internal reference protein. Band density data were used to draw the graph. (**F**) The migration of MDA-MB-231 cells treated with or without 13*R*,20-diHDHA (RPMI1640/0.5% FBS) was imaged at 0, 6, and 12 h by a scratch assay (scale bar: 100 μm), and the area was calculated using the Image J software. * *p* < 0.05 versus the DMSO-treated control group. (**G**) The cell migration (without Matrigel) and invasion (with Matrigel) of MDA-MB-231 cells exposed to 13*R*,20-diHDHA were determined by transwell assays (scale bar: 100 μm). (**H**) Colony formation assays were performed on MDA-MB-231 and MCF-7 cells that had been incubated in 6-well plates and treated with 13*R*,20-diHDHA (20 μM). Representative colony formation data were collected. The data from triplicate experiments are presented as the mean ± SD.

**Figure 3 antioxidants-10-00457-f003:**
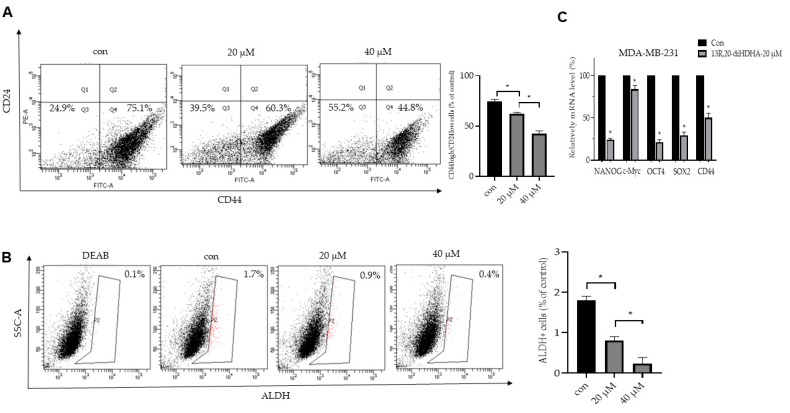
The effect of 13R,20-diHDHA on the CD44^high^/CD24^low^ and aldehyde dehydrogenase (ALDH)-positive cell proportions. (**A**) The CD44^high^/CD24^low^ cell populations of MDA-MB-231 cells treated with 13R,20-diHDHA (20 or 40 μM) or DMSO for 24 h were analyzed by FACS. The gating was based on binding of a control antibody. (**B**) 13R,20-diHDHA decreased the ALDH-positive cell population, as detected with an ALDEFLUOR™ kit (Vancouver, BC, Canada). Breast cancer cells were treated with 13R,20-diHDHA (20 or 40 μM) for 24 h and subjected to FACS analysis. Representative flow cytometric data are shown. The left panel shows the ALDH-positive population in the presence of the ALDH inhibitor, DEAB, and the right panel represents the ALDH-positive population without DEAB. (**C**) Transcript levels of the CSC markers, Nanog, Sox2, Oct4, c-Myc, and CD44, were determined in 13R,20-diHDHA-treated mammospheres using gene-specific primers and real-time RT-qPCR. β-actin was detected as an internal control. The data from triplicate experiments are presented as the mean ± SD. * *p* < 0.05 versus the DMSO-treated control group.

**Figure 4 antioxidants-10-00457-f004:**
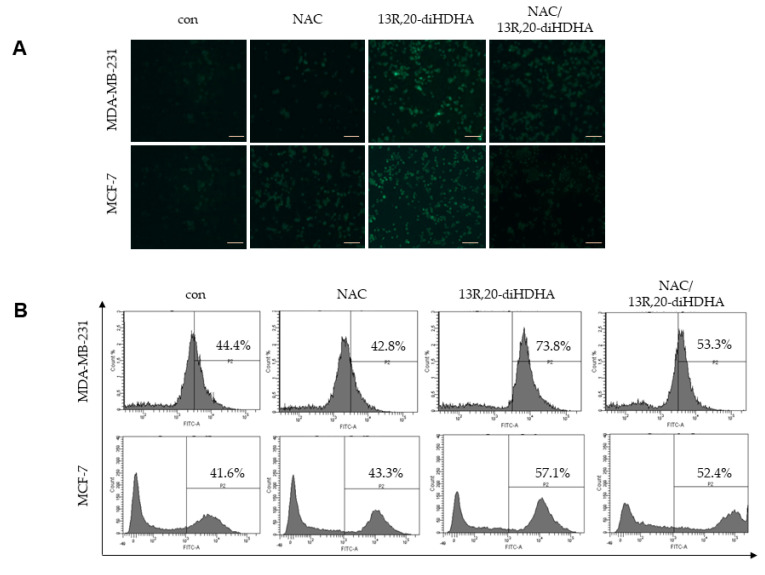
Effect of 13*R*,20-diHDHA-induced ROS generation on mammospheres formation. (**A**,**B**) The effect of 13*R*,20-diHDHA (20 μM) on ROS generation in MDA-MB-231 and MCF-7 cells was determined using DCF-DA staining. Images were obtained by FACS and fluorescence microscopy at 4x magnification and representative photos are shown (scale bar: 100 μm). (**C**) Mammospheres were pretreated with/without NAC (10 mM) for 1 h prior to treatment with 20 μM 13*R*,20-diHDHA. After 7 days, mammospheres formation was determined. Representative images of colonies were recorded. The data from triplicate experiments are presented as the mean ± SD. * *p* < 0.05 versus the DMSO-treated control group.

**Figure 5 antioxidants-10-00457-f005:**
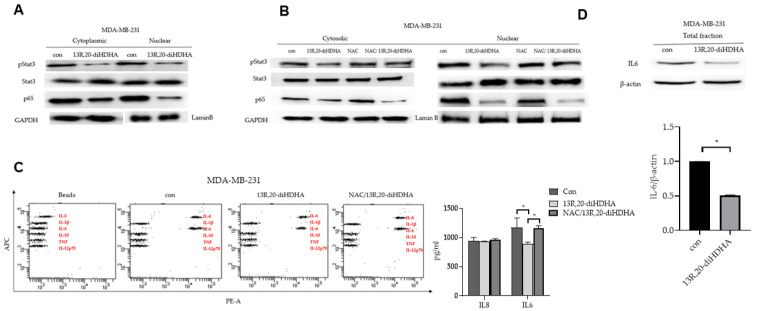
Effects of 13*R*,20-diHDHA on Stat3 pathway activation in mammospheres and CSC loads in breast cancer. (**A**) The activation of Stat3 and NF-κB was determined in mammospheres using antibodies against pStat3, Stat3, p65, and GAPDH, Lamin B. 13*R*,20-diHDHA decreased the nuclear levels of pStat3 and p65 in the mammospheres. (**B**) The effects of 13R,20-diHDHA and NAC on pStat3 phosphorylation. The 13*R*,20-diHDHA-induced dephosphorylation of pStat3 was ameliorated by NAC. (**C**) Cytokine profile assay of the conditioned media of control and 13*R*,20-diHDHA-treated cells, as performed using specific antibodies and cytokine beads. The quantification of cytokines already been done using by BD CBA cytokine assay kit and FCAP Array program. (**D**) Mammospheres culture cells were immunoblotted with antibodies against IL-6 and β-actin (internal reference protein). The data from triplicate experiments are presented as the mean ± SD. * *p* < 0.05 versus the DMSO-treated control group.

**Table 1 antioxidants-10-00457-t001:** Specific Real-time RT-qPCR primers sequences containing human and β-actin genes.

Genes	Primers
CD 44	Forward: 5′-AGAAGGTGTGGGCAGAAGAA-3′
	Reverse: 5′-AAATGCACCATTTCCTGAGA-3′
NANOG	Forward: 5′-ATGCCTCACACGGAGACTGT-3′
	Reverse: 5′-AAGTGGGTTGTTTGCCTTTG-3′
OCT4	Forward: 5′-AGCAAAACCCGGAGGAGT-3′
	Reverse: 5′-CC ACATCGGCCTGTGTATATC-3′
SOX2	Forward: 5′-TTGCTGCCTCTTTAAGACTAGGA-3′
	Reverse: 5′-CTGGGGCTCAAACTTCTCTC-3′
c-Myc	Forward: 5′-AATGAAAAGGCCCCCAAGGTAGTTATCC-3′
	Reverse: 5′-AGCAAAAC CCGGAGGAGT-3′
β-actin	Forward: 5′-TGTTACCAACCTGGGACGACA-3′ Reverse: 5′-GGGGTGTTGAAGGTCTCAAA-3′

## Data Availability

The data presented in this study are available on request from the corresponding author.

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
