# Peer review of "13R,20-Dihydroxydocosahexaenoic Acid, a Novel Dihydroxy- DHA Derivative, Inhibits Breast Cancer Stemness through Regulation of the Stat3/IL-6 Signaling Pathway by Inducing ROS Production"

_antioxidants, 2021, doi:10.3390/antiox10030457_

Round 1

Reviewer 1 Report

  • Figure 4 is extremely confusing. It is unclear why the authors chose to study the NADPH oxidase complex, which is an extracellular facing membrane protein, while, the intent of the others is to study the ROS production. I would think it’s more logical to study mitochondrial NADH (and not NADPH) to corroborate their ROS data findings. Alternatively, if the authors wanted to study NADPH oxidase, do the authors mean to say that their compound inhibits phagosome functionality of cancer cells. By the way, the phagosome functional of cancer cell, in itself, does not make much sense. It seems the authors have confused or atleast confused the reviewer between NADH oxidase and NADPH oxidase. Line 422 should be NADPH. Intracellular ROS is better studied by superoxide dismutase (SOD-1/2) functionality. I wonder why authors did not study the expression of SOD in the context of their drug-mediated alterations in ROS. Importantly, the title line (#422) says NADH, but data is on NADPH oxidase.
  • As 13R, 20-diHDHA is obtained from DHA, and DHA in itself is a known anti-inflammatory compound, why is it the authors did not use DHA or 14R,21-isomeric form as controls to see if 13R,20-form performed better. Authors should atleast provide cytokine data with DHA and 14R,21-form treatment and compare it with 13R, 20-form.
  • If the compound only effected ~1% of cancer stem cells, what is the logic behind figure 2? Why did the authors' test for cell migration? Cell migration assay made me think of metastasis? This made me think of the impact of the compound on VEGF-A expression or CXCR4 chemokine receptor expression. This does not seem to be the focus of the paper. So either the authors should explore VEGF-A and CXCR4 (or any related migratory factor) expression or remove these figures as it does not add value to their current study.

Minor comments:

Line 307 to 311 complex and needs to be rephrased.

Line 399 4T1 is repeated

Statistical analysis for figure 3 is unclear.

Author Response

Dear Reviewer:

Thank you for your letter and comments concerning our manuscript entitled “13R,20-dihydroxy Docosahexaenoic Acid, a Novel DHA Dihydroxy Derivative, Inhibits Breast Cancer Stemness through Regulation of Stat3/IL-6 Signaling Pathway by Inducing the ROS Production”. (Antioxidants-1096985)

We greatly appreciate your positive comments. Those comments are all valuable and very helpful for revising and improving our paper, as well as the important guiding significance to our researches. We have studied comments carefully and have made correction which we hope to meet with approval. Revised part is marked in red in the paper. The main corrections in the paper and the response to your comments are as in attached file.

Reviewer 2 Report

Strengths:
In this manuscript, the authors reported the outcomes of their study in which they investigated the effects of 13R,20-dHDHA on macrophage IL-6 and TNF-alpha secretion. More importantly, they investigated the effects of 13R,20-dHDHA on breast cancer stemness under various experimental conditions. They also examined the effect of this compound on ROS levels and STAT3 phosphorylation. They observed that 13R,20-dHDHA exhibited dose-dependent inhibition of mammospheres formation, colony formation, migration and invasion. They also observed that 13R,20-dHDHA reduced markers and genes associated with breast cancer stemness. In addition, they observed that 13R,20-dHDHA increased ROS production in the cells via NADPH oxidase activation, enhanced nuclear NF-kB activity and IL-6 secretion but significantly reduced STAT3 phosphorylation. From these findings, the authors concluded that 13R,20-dHDHA inhibits breast cancer stemness through ROS production and downstream downregulation of STAT3/IL-6 signaling. Based on these findings, the authors suggested that 13R,20-dHDHA has anti-cancer effects and should be considered for anti-cancer therapy development.

Previously the authors had synthesized and characterized 13R,20-dHDHA including its purity. Obviously, in their current study whose results are reported here, the authors have provided hosts of strong and related evidence that point to potential anti-cancer effects of this compound in breast cancer stemness. In addition, the authors provided some mechanistic insight into how their compound inhibits IL-6 secretion and breast cancer stemness. Overall, the study designs were excellent and meet the scientific standards.

Their data collection, presentation and interpretation are all of high quality and meet the standard of this journal. The authors wrote the manuscript in very good grammar that is easy to read and follow. The conclusions are well supported by their results. Based on aforementioned facts, I recommend that you give the authors opportunity for slight revision to include their responses to the questions below as well as include citations suggested below, which would further strengthen the discussion section and the overall merit of the manuscript.       

Concerns to be addressed:

1. The authors did not provide any indication or evidence regarding how 13R,20-dHDHA causes reduction in STAT3 phosphorylation. Since the authors didn’t determine the effect of 13R,20-dHDHA on Jak2, it is impossible to know whether the reduction of STAT3 phosphorylation they observed was caused by inhibition of Jak2 by their compound. There are endogenous protein inhibitors of Jak kinases, which include SOCS1 or SOCS3. It is possible that 13R,20-dHDHA activates either SOCS1 or SOCS3, which directly inhibits Jak2, thus leading to reduced phosphorylation of STAT3. These possibility should be included in the discussion.

2. The decline in STAT3 phosphorylation induced by their compound could also have been caused by activation of a protein tyrosine phosphatase such as SHIP1 or SHIP2, which are known to dephosphorylate STAT3 resulting in negative regulation. Thus, it is also possible that their compound activated SHIP1 or SHIP2, which dephosphorylated STAT3 downstream of the IL-6R activation. This should be included in their discussions. Please, refer to Mihwa Kim et al. J. Mol. Sci.2018 Sept 11;19(9):2708. Doi:10.3390/ijms190092708

3. In 2011, Kyung D, et al reported on “Potential role of NADPH oxidase-mediated activation of Jak2 leads to STAT3 activation". Please, refer to Kyung D, et al in 2011 paper. It appeared they observed NADPH oxidase mediated activation of Jak leading to STAT3 phosphorylation. But, in this manuscript you showed that NADPH oxidase induces STAT3 dephosphorylation in cells treated with your compound. This means the role of NADPH you reported here is somehow contradictory to the role of NADPH oxidase regarding STAT3 regulation.  Please, refer to Kyung's paper and make the appropriate comment on it in the discussion section of your manuscript.  

4. After treating the THP-1 cells with PMA to differentiate them to macrophage, did you check to see what percentage of the original THP-1 cells were converted to macrophages before treating them with 13R,20-dHDHA? This is important to make sure you had largely macrophage population for the experiments.

5. Please, correct a typo error (abovemmentioned) on line 261.

Author Response

Dear Reviewer:

Thank you for your letter and comments concerning our manuscript entitled “13R,20-dihydroxy Docosahexaenoic Acid, a Novel DHA Dihydroxy Derivative, Inhibits Breast Cancer Stemness through Regulation of Stat3/IL-6 Signaling Pathway by Inducing the ROS Production”. (Antioxidants-1096985)

We greatly appreciate your positive comments. Those comments are all valuable and very helpful for revising and improving our paper, as well as the important guiding significance to our researches. We have studied comments carefully and have made correction which we hope meet with approval. Revised portion are marked in red in the paper. The main corrections in the paper and the responds to your comments are as attached file.

Reviewer 3 Report

The work by Wang et al analyzes the putative antitumoral effect of a novel dihydroxy DHA derivative on two different breast cancer cell lines. First, authors check the anti-inflammatory activity of the compound by measuring the inhibition of LPS-induced IL6 and TNFa release by THP1 cells differentiated to macrophages. The compound reduced mamosphere and colony formation, migration and invasion, as well as CD44high/CD24low and ALDH+ populations. They also show the compound induces ROS production, which is reverted with the antioxidant NAC, recovering full mamosphere formation ability with the double treatment. Moreover, the link with inflammation-related cellular signaling (stat3 and NFkB) was checked by western blot. Finally, authors show the compound diminished pluripotency-related gene expression and proliferation of mamosphere-derived cells.

Major comments:

-Experiments across the manuscript need to show results for both cell lines: Figure 2 (B, C, E, F), Figure 3, Figure 4 (C,D), Figure 5 (at least A, B, D), Figure 6. Indeed, in some panels the figure legend indicates both cell lines but the figure shows only one (i.e. 2F) and many of them doesn´t even indicate in the figure legend or the text the cell line shown.

-DPI is a flavoprotein inhibitor not specific for NADPH oxidases, it can also inhibit mitochondrial ROS. I would advise to repeat the experiment in Figure 4D with alternative inhibitors or remove the link with NADPH oxidases altogether. Importantly, the effect of the inhibitor on ROS production needs to be shown to reach the conclusion in lines 427-428:  “Figure 4D shows that 5 μM DPI attenuated the ability of 13R,20-diHDHA-induced ROS to inhibit mammospheres formation”.

-Western blots in Figure 5A, B need to show both cytosolic and nuclear fractions to distinguish between phosphorylation and translocation. Additionally, 5C need to show IL6 levels in response to NAC+compound to be able to conclude that cytokine release is dependent on ROS production.

-The sentence: “Together, these results show that 13R,20-diHDHA-induced ROS inhibits breast cancer stemness via Stat3/IL-6 signaling, and that 13R,20-diHDHA directly inhibits cancer cell stemness via NF-κB pathways” (lines 447-449) is a clear overstatement, please tone down. To demonstrate such direct relationships, experiments with inhibitors/knockdown of stat3 and NFkB would need to be included to directly link stemness and the effects of the compound with these signaling pathways. The results shown only demonstrate that the compound reduced the level of both p- stat3 and p65 in the nucleus, and that the first one is dependent on ROS.

-Results shown if Figure 6 would be better placed together with Figures 2, 3 since they describe effects of the compound on stemness.

Minor comments:

-length of the introduction needs to be reduced, it seems too extensive to me

-Figure 1B: please, complete labeling of the pictures: what´s the difference between the ones in the right and the ones in the left? Also, visible scale bars need to be shown.

-FACS plots in Figure 3A look off: there are cells below the scale for CD24.

-Figure 5C: How is quantification of cytokines done? Can authors show the values of mean fluorescence intensity of the PE channel for IL8 and IL6 in histograms? Probably it would be much more informative and it would facilitate understanding of the figure.

-Two of the authors do not have any contribution assigned

-Western blots shown in supplementary figure are far from being “uncropped” gels. In fact, in the last slide, we can guess the lines corresponding to molecular weights, which do not correspond to the arrows shown by the authors. Please, completely labelled gels need to be shown.

Author Response

Dear Reviewer:

Thank you for your letter and comments concerning our manuscript entitled “13R,20-dihydroxy Docosahexaenoic Acid, a Novel DHA Dihydroxy Derivative, Inhibits Breast Cancer Stemness through Regulation of Stat3/IL-6 Signaling Pathway by Inducing the ROS Production”. (Antioxidants-1096985)

We greatly appreciate your positive comments. Those comments are all valuable and very helpful for revising and improving our paper, as well as the important guiding significance to our researches. We have studied comments carefully and have made correction which we hope meet with approval. Revised portion are marked in red in the paper. The main corrections in the paper and the responds to your comments are as in attached file.

Reviewer 4 Report

In their manuscript, Wang and co-workers investigated the anti-inflammatory effect of 13R,20-dihydroxy docosahex-16 aenoic acid a novel DHA dihydroxy derivative, which  has been descovered and studied by the same authors and reported in a previously published manuscript. In this manuscript the authors reports that 13R,20-diHDHA reduced the macrophage secretion of  IL-6 and TNF-α. Moreover, the authors investigate the effect of the new resolvin in breast cancer cell lines. 

To this reviewer the findings should be inetersting and releventa but the results and conclusion needs some improvements. My most concerns are on the structure of the results: previous figures are on THP1 cells differentiated to macrophages and the others are on the new HDHA effects on cancer stem cells. I STRONGLY suggest co-culture experiments to demonstrate the relation between TME and cancer cells. This will support also the conclusions, which are  overexstimated.  

Author Response

(The authors gave the same response as above.)

Round 2

Reviewer 1 Report

No further comments.

Author Response

Dear Reviewer:

Thank you for your letter and comments concerning our manuscript entitled “13R,20-dihydroxy Docosahexaenoic Acid, a Novel DHA Dihydroxy Derivative, Inhibits Breast Cancer Stemness through Regulation of Stat3/IL-6 Signaling Pathway by Inducing the ROS Production”. (Antioxidants-1096985). We greatly appreciate your positive comments. 

Reviewer 3 Report

The manuscript has been improved in some aspects, but many of my concerns remain unanswered.

  1. As I commented in my previous report, DPI is a flavoprotein inhibitor not specific for NADPH oxidases, it can also inhibit mitochondrial ROS. I would advise to repeat the experiment in Figure 4D with alternative inhibitors for NADPH oxidases, comment jointly the results with NAC and DPI or remove the link with NADPH oxidases altogether.
  2. Western blots in Figure 5A, B need to show both cytosolic and nuclear fractions to distinguish between phosphorylation and translocation. Additionally, 5C need to show IL6 levels in response to NAC+compound to be able to conclude that cytokine release is dependent on ROS production.
  3. The corrected sentence: “Together, these results show that 13R,20-diHDHA-induced ROS inhibits breast cancer stemness via Stat3/IL-6 signaling.” (lines 397-398) is still an overstatement, please tone down since results shown do not demonstrate such a direct link between ROS and IL6.
  4. Figure 1B: please, complete labeling of the pictures: what´s the difference between the ones in the right and the ones in the left?
  5. Western blots shown in supplementary figure are far from being “uncropped” gels, they need to show the complete gel.

Author Response

Dear Reviewer:

Thank you for your letter and comments concerning our manuscript entitled “13R,20-dihydroxy Docosahexaenoic Acid, a Novel DHA Dihydroxy Derivative, Inhibits Breast Cancer Stemness through Regulation of Stat3/IL-6 Signaling Pathway by Inducing the ROS Production”. (Antioxidants-1096985)

We greatly appreciate your positive comments. Those comments are all valuable and very helpful for revising and improving our paper, as well as the important guiding significance to our researches. We have studied comments carefully and have made correction which we hope meet with approval. Revised portion are marked in red in the paper. The main corrections in the paper and the responcese to your comments are in the attachment file.

Reviewer 4 Report

Dear Authors, i send you the same comment that i'll send to the Editor. The paper has been well reviewed BUT, as i comunicate in my REV1, i think that co-culture experiments are fundamental. So please, take the time to perform experiments, insert the results and the paper will be significant for the field. 

Author Response

Dear Reviewer:

Thank you for your letter and comments concerning our manuscript entitled “13R,20-dihydroxy Docosahexaenoic Acid, a Novel DHA Dihydroxy Derivative, Inhibits Breast Cancer Stemness through Regulation of Stat3/IL-6 Signaling Pathway by Inducing the ROS Production”. (Antioxidants-1096985)

We greatly appreciate your positive comments. Those comments are all valuable and very helpful for revising and improving our paper, as well as the important guiding significance to our researches. We have studied comments carefully and have made correction which we hope meet with approval. Revised portion are marked in red in the paper. The main corrections in the paper and the responds to your comments are as in the attachment file.

Round 3

Reviewer 3 Report

No further comments